# Activity of *N*-Chlorotaurine against Long-Term Biofilms of Bacteria and Yeasts

**DOI:** 10.3390/antibiotics10080891

**Published:** 2021-07-22

**Authors:** Victoria Grimus, Débora C. Coraça-Huber, Stephan J. M. Steixner, Markus Nagl

**Affiliations:** Research Laboratory for Biofilms and Implant Associated Infections (BIOFILM LAB), Institute of Hygiene and Medical Microbiology, University Hospital for Orthopaedics and Traumatology, Medical University of Innsbruck, A-6020 Innsbruck, Austria; victoria.grimus@student-i-med.ac.at (V.G.); debora.coraca-huber@i-med.ac.at (D.C.C.-H.); stephan.steixner@i-med.ac.at (S.J.M.S.)

**Keywords:** *N*-chlorotaurine, biofilm, antiseptic, microbicidal, fungicidal, bactericidal

## Abstract

*Background: N*-chlorotaurine (NCT), an antiseptic that originates from the human defense system, has broad-spectrum microbicidal activity and is well tolerated by human tissue and applicable to sensitive body regions. Bacteria in short-term biofilms, too, have been shown to be killed by NCT. It was the aim of the present study to demonstrate the activity of NCT against bacteria and yeasts in longer-lasting biofilms, including their co-culture. *Materials and methods: Staphylococcus aureus*, *Pseudomonas aeruginosa*, and *Klebsiella variicola* biofilms were grown for 14 weeks in MBEC^TM^ inoculator with 96 well base. Some pegs were pinched off weekly and incubated in 1% NCT in PBS (PBS only for controls) at pH 7.1 and 37 °C, for 30 and 60 min. Subsequently, bacteria were resuspended by ultrasonication and subjected to quantitative cultures. Similar tests were conducted with *C. albicans* biofilms grown on metal (A2-steel) discs for 4 weeks. Mixed co-cultures of *C. albicans* plus each of the three bacterial strains on metal discs were grown for 5–7 weeks and weekly evaluated, as mentioned above. *Results:* Single biofilms of *S. aureus*, *P. aeruginosa*, and *K. variicola* grew to approximately 1 × 10^6^ colony forming units (CFU)/mL and *C. albicans* to 1 × 10^5^ CFU/mL. In combined biofilms, the CFU count was about 1 log_10_ lower. Viable counts of biofilms of single bacteria were reduced by 2.8 to 5.6 log_10_ in 1% NCT after 60 min (0.9 to 4.7 log_10_ after 30 min) with Gram-negative bacteria being more susceptible than *S. aureus*. Significant reduction of *C. albicans* by 2.0 to 2.9 log_10_ occurred after 4 h incubation. In combined biofilms, viable counts of *C. albicans* were reduced by 1.1 to 2.4 log_10_ after 4 h, while they reached the detection limit after 1 to 2 h with bacteria (2.0 to > 3.5 log_10_ reduction). Remarkably, older biofilms demonstrated no increase in resistance but constant susceptibility to NCT. This was valid for all tested pathogens. In electron microscopy, morphological differences between NCT-treated and non-treated biofilms could be found. *Conclusions:* NCT is active against long-term biofilms of up to several months irrespective of their age. Combined biofilm cultures of yeasts and bacteria show a similar susceptibility pattern to NCT as single ones. These results contribute to the explanation of the clinical efficacy of NCT, for instance, in infected chronic wounds and purulently coated crural ulcerations.

## 1. Introduction

Biofilms are a long-lived, organized community of microorganisms within an extracellular matrix. Biofilm-related infections count as nosocomial infections and are a major cause of death and increased morbidity among hospitalized patients. These infections increase the pressure on medical systems already overwhelmed, in both developed and undeveloped countries [1]. Because of the rising number of patients with this serious complication, the orthopedic community, for example, has become more concerned about the management of implant infections. Those infections are difficult to treat often requiring surgical implant replacement [2]. The most commonly isolated microorganisms in implant infections are Coagulase-Negative *Staphylococci* (CoNS; primarily *S. epidermidis*), followed by *S. aureus* and mixed flora [3,4,5]. In previous decades, resistance of CoNS and several other microorganisms against antibiotics increased. Strong and sometimes dramatic increases in the percentage of resistant isolates were noted particularly for penicillin, oxacillin, ciprofloxacin, clindamycin, erythromycin, and gentamicin [6]. The ability of these bacteria to produce biofilms is an additional mechanism contributing to this phenomenon and negatively affecting the antimicrobial susceptibility of CoNS [7]. Biofilm formation also explains why some normal flora organisms traditionally considered “harmless” become pathogenic when they grow on the surface of foreign bodies [8].

*N*-chlorotaurine (NCT) is a long-lived oxidant produced by activated human granulocytes and monocytes during the oxidative burst [9,10]. As a sodium salt, it can be used as a mild, but effective, and well tolerated antiseptic and anti-infective for topical treatment of infections of different body sites, for instance, the skin and mucous membranes, the eye, the ear, and the urinary bladder [11,12]. In the last years, also the lower airways became a topic of interest, since inhalation of NCT turned out to be well tolerated [13]. Therefore, acute bronchopulmonary infections including the recent COVID-19 pandemic and chronic diseases such as chronic obstructive pulmonary disease and cystic fibrosis may become indications for NCT treatment [13,14,15].

NCT has demonstrated microbicidal activity against bacteria, viruses, fungi, protozoa, and worm larvae [11]. Viability of biofilms of *Staphylococcus aureus*, *Staphylococcus epidermidis*, and *Pseudomonas aeruginosa* is largely reduced by the preferred pharmacological concentrations of 1% (55 mM) NCT, too, after incubation times of 15 min to 1 h [16,17]. Lower millimolar NCT concentrations kill the biofilm at longer incubation times [16,17], and 300 µM inhibited the formation of biofilms [18]. In clinical studies, NCT has shown significant curative effects in infections or disorders where biofilms are thought to play a role, namely in purulently coated leg ulcers [19], external otitis [20], and dental plaque [21]. In such chronic diseases, persisting biofilms appear to be causative for antibiotic resistance and therapeutic problems. A previous study demonstrated increasing resistance of biofilms grown for 2–3 weeks to antiseptics (hypochlorite, iodine, and chlorhexidine) with no further change of susceptibility up to 8 weeks [22]. The activity of NCT against long-term biofilms is unknown.

Because of the encouraging clinical results, we started to focus on the activity of NCT against biofilms grown over longer time-periods. Using the model of MBEC plates, it was possible to monitor the susceptibility of biofilms to NCT for 3 to 5 months. Another unknown aspect was addressed in this study, which is the activity of NCT against combined biofilms of bacteria and fungi.

## 2. Materials and Methods

### 2.1. Bacterial and Fungal Strains

*S. aureus* ATCC 25923 and ATCC 6538, *P. aeruginosa* ATCC 27853, *Klebsiella variicola* (clinical isolate), and *Candida albicans* strains ATCC 15053, ATCC 5314, and CBS 5982 were used. *S. aureus*, *P. aeruginosa*, and *C. albicans* were chosen as pathogens frequently isolated from human samples, while *K. variicola* is an emerging opportunistic pathogen widespread in humans and animals with the potential of hypervirulence and biofilm formation, too [23,24]. Bacteria and yeast, deep frozen at minus 80 °C for storage, were thawed and grown on Mueller–Hinton agar plates (Oxoid Ltd., Oxoid, UK) for use. Overnight cultures from the plates were grown at 37 °C in tryptic soy broth (Merck, Darmstadt, Germany) to 3–5 × 10^9^ colony forming units (CFU)/mL for bacteria and grown in tryptic soy broth plus 1% glucose to 1 × 10^7^ CFU/mL for yeasts, respectively.

### 2.2. N-Chlorotaurine

Pure NCT was prepared as crystalline sodium salt (Cl–HN–CH_2_–CH_2_–SO_3_Na; molecular weight, 181.57 [25]) in pharmaceutical quality and dissolved in 0.1 M phosphate buffer (pH 7.1) to a concentration of 1% (55 mM). This concentration turned out as a standard clinically tolerated in different body sites and demonstrating good microbicidal activity against all kinds of pathogens [11,12,26].

### 2.3. Devices for Growth of Biofilm

Two devices were applied. The first was MBEC™ biofilm inoculator with 96 well base (No. 19112 MBEC P&G Panel, Innovotech Inc., Edmonton, AB, Canada). The biofilm grows on pegs fixed to the lid and soaked in the wells. For evaluation, the pegs were snapped off and subjected to ultrasonication and quantitative cultures or to electron microscopy (for details, please see below).

The second device was metal discs, A2-steel, DIN 9021, size M2, external diameter 6 mm, internal diameter 2.2 mm, thickness 0.8 mm (Schrauben-Niro.de, SyneriaX, Bramstedt, Germany). The metal discs were put into 96-well flat bottom microtitre plates (Corning, NY, USA) for growth of biofilm (for detailed procedure, please see below).

### 2.4. Growth of Biofilm of Bacteria in MBEC Inoculator and Incubation in NCT

Aliquots of overnight cultures of the pathogens were 100-fold diluted in tryptic soy broth, and 150 µL of the dilutions each were filled into the wells of the 96-well base of the inoculator. All bacteria were tested separately. The initial concentration of bacteria was 3–5 × 10^7^ colony forming units (CFU)/mL. The lid with the pegs was put on the base, and the device was laid in a plastic chamber with a wet towel to warrant a moist condition. The whole chamber was incubated in a rotatory shaker at 150 rpm and 37 °C for 11–21 weeks for *P. aeruginosa*, for 11–19 weeks for *S. aureus* ATCC 25923, and for 14–19 weeks for *K. variicola*. The medium (tryptic soy broth) was changed twice a week each in all wells.

Tests for susceptibility to NCT were performed weekly. For this, six pegs each were pinched off with forceps and washed in 2 mL PBS in wells of a 24-well plate (Corning, NY, USA). Subsequently, four pegs were transferred into wells containing 2 mL of 1% NCT in PBS (pH 7.1) and two pegs in wells with 2 mL of PBS only for controls, one peg per well. Two of the test pegs were incubated for 30 min, and two for 60 min in NCT at 37 °C. Control pegs were incubated for 60 min. After the incubation time, the pegs were washed twice in PBS. One peg was subjected to quantitative colony counts, the other one to scanning electron microscopy.

### 2.5. Growth of Biofilm of Yeasts and Mixed Cultures of Yeast and Single Bacteria on Metal Discs and Incubation in NCT

Aliquots of overnight cultures of *Candida spp*. were 100-fold diluted in tryptic soy broth plus 1% glucose, and 150 µL of the dilutions each were filled into the wells of 96-well flat bottom microtiter plates. The wells of the plates were filled with one metal disc each. All yeast strains were tested separately. The initial concentration of yeast was approximately 1 × 10^5^ colony forming units (CFU)/mL. The plates were put in a moist chamber and agitated at 200 rpm at 37 °C, as mentioned, above for 4 weeks. Medium changes were performed twice a week. Tests for susceptibility to NCT were performed weekly. The procedure was similar to the one with the pegs described in the previous paragraph with the only difference that metal discs were used instead of pegs and that 48-well plates were filled with 1 mL of NCT or PBS for the test and control incubation and washing steps.

For mixed cultures, *C. albicans* ATCC 5314 and the bacteria were grown separately overnight in tryptic soy broth plus 1% glucose. Subsequently, *C. albicans* was 100-fold diluted in tryptic soy broth plus 1% glucose, while *S. aureus*, *P. aeruginosa*, and *K. variicola* were 10,000-fold diluted in 0.9% saline. For the mixed biofilms, *S. aureus* strain ATCC 6538 was used because of its yellow colony color, which renders it easily distinguishable from *C. albicans* on agar plates. *C. albicans* was tested with each strain of the bacteria separately in a mixed culture. Aliquots of 0.2 mL of *C. albicans* and 0.2 mL of each bacterial strain were diluted in 19.6 mL of tryptic soy broth plus 1% glucose. Therefore, the final concentration of the yeast was approximately 1 × 10^3^ CFU/mL, and that of the bacteria approximately 3 × 10^3^ CFU/mL. Again, 150 µL of the dilutions each were filled into the wells of 96-well flat bottom microtiter plates, which contained one metal disc per well. The plates were put in a moist chamber and agitated at 200 rpm at 37 °C, as mentioned above, for 5 to 7 weeks. The pattern of medium changes and susceptibility tests was similar to the tests with *C. albicans* single biofilm cultures.

### 2.6. Quantitative Cultures

After incubation in NCT or control PBS, the washed pegs and metal disks, respectively, were transferred into 15 mL Falcon tubes containing 1 mL of PBS. They were vortexed three times for 5 s, sonicated for 1 min in an ultrasound water bath (40 kHz; BactoSonic; Bandelin Electronic, Berlin, Germany), and vortexed again three times to detach the remaining live bacteria from the pegs or disks. Test samples were processed undiluted and 10-fold diluted, and controls were 100-fold diluted. Aliquots of 50 μL of these solutions were spread on Mueller–Hinton agar plates in duplicate, using an automatic spiral plater (model WASP 2; Don Whitley, Shipley, West Yorkshire, UK). The detection limit was 10 CFU/mL, taking into account both plates. Plates were incubated for 48 h (bacteria) to 72 h (fungi) at 37 °C, and the number of CFU was counted.

### 2.7. Scanning Electron Microscopy

A procedure was used similar to a previous study on the impact of NCT on short-term biofilms [17]. After incubation in NCT or control PBS, the washed pegs and metal disks, respectively, were fixed with 2.5% glutaraldehyde (BioChemika Fluka, Buchs, Switzerland) in 0.1 M phosphate buffer (pH 7.4). After a brief wash in phosphate buffer, the samples were gradually dehydrated with ethanol. After the last step, the discs were placed in an incubator for drying out. The dried discs were placed on aluminum pins and fixed with Leit-C (Göcke, Plano GmbH, Wetzlar, Hessen, Germany). The pins were sputtered with Au 10 nm (Agar Sputter Coater, Agar Scientific Ltd., Stansted, GB, UK) for 1 min and analyzed by scanning electron microscopy (SEM, JSM-6010LV, JEOL GmbH, Freising, Germany).

### 2.8. Statistics

Several independent repetition experiments were performed for each condition. The results are presented as mean values and standard deviation or standard error of the mean, as indicated in the results. One-way analysis of variance (ANOVA) and Dunnett’s and Tukey’s multiple comparison tests were carried out for comparison of test samples with controls and for different incubation times in NCT, respectively. *p* values < 0.05 were considered significant.

## 3. Results

Biofilms developed well and could be kept for many weeks both in the MBEC™ biofilm inoculator and on A2-steel discs. The limiting factor is contamination with other pathogens despite strict aseptic workflow, but several independent repetition experiments over 3.5 months were possible. Single tests reached 21 weeks for *P. aeruginosa* and 19 weeks for *S. aureus* and *K. variicola*, with similar results and no additional information. Overall, the aspect of the biofilm in electron microscopy did not change significantly over the months, and the susceptibility to NCT remained constant.

### 3.1. Bactericidal Effect of NCT in Long-Term Biofilms of Single Strains

These biofilms were grown in MBEC inoculators at 37 °C for 14 weeks. Bacterial counts reached about 1 × 10^6^ CFU/mL. Overall, the standard concentration of 1% NCT showed bactericidal activity after 30 min incubation time at 37 °C and pH 7.1, which increased after 60 min. The course of susceptibility over 14 weeks is shown in Figure 1, Figure 2 and Figure 3 for *S. aureus* ATCC 25923, *P. aeruginosa* ATCC 27853, and *K. variicola* (clinical isolate). *S. aureus* was less susceptible than the Gram-negative bacteria. Single time points of evaluation did not reach significance for killing after 30 min incubation in NCT because of high standard deviations. Note the presentation with standard error of the mean in the figures. The 30 min incubation of *S. aureus* biofilm in NCT may be suggestive of decreasing activity with increasing age of the biofilm, but this was insignificant as well. The Gram-negative bacteria had a higher susceptibility with some (*P. aeruginosa*) or all (*K. variicola*) single values significant after 30 min incubation and with a reduction of viable counts nearly or completely to the detection limit of 1 log_10_ after 60 min incubation.

The absence of a change of susceptibility over time allowed a calculation over all single values, which is shown in Figure 4. The overall reduction in CFU was highly significant for 30 and 60 min NCT for all strains, coming to 0.91 and 2.83 for *S. aureus*, 2.27 and 5.13 for *P. aeruginosa*, and 4.74 and 5.58 for *K. variicola*, respectively (Figure 4).

### 3.2. Fungicidal Effect of NCT in Long-Term Biofilms of Single C. albicans Strains

These biofilms were grown on A2-steel discs at 37 °C for 4 weeks. Biofilms of all yeast test strains grew to approximately 1 × 10^5^ CFU/mL and were similarly susceptible to NCT. As with bacteria, there was no change in the CFU counts and in the susceptibility to NCT over the whole test period. As an example, the detailed course is shown in Figure 5a for *C. albicans* ATCC 5314. The summary of the values similar to Figure 4 allowed a good overview and disclosed the longer incubation times of 3 to 4 h in 1% NCT at 37 °C and pH 7.1 needed for significant killing of yeasts compared to bacteria (Figure 5b). The extent of killing was similar using the *C. albicans* strains ATCC 15053 and CBS 5982 (Figure 6). A reduction in CFU by >3 log_10_ nearly to the detection limit could be achieved after 5 h incubation in 1% NCT. For detailed values, see Figure 5 and Figure 6.

### 3.3. Bactericidal and Fungicidal Effect of NCT in Mixed Long-Term Biofilms

These biofilms were grown on A2-steel discs at 37 °C for 5 (*S. aureus* ATCC 6538) to 7 weeks (*P. aeruginosa* and *K. variicola*). *C. albicans* ATCC 5314 was chosen as the partner for each bacterial strain. The yeast grew to lower colony counts in mixed cultures than in single cultures, coming to roughly 1 × 10^4^ CFU/mL together with *P. aeruginosa* and 1x 10^3^ CFU/mL with *S. aureus* and *K. variicola*. Moreover, *S. aureus* and *K. variicola* achieved lower counts by approximately 1 log_10_ than in single cultures, although this is not directly comparable because of the different biofilm culture systems.

Again, all microorganisms in mixed biofilm cultures were killed by 1% NCT at 37 °C and pH 7.1. The reduction rate of all was similar to that of single strain cultures. The age of the biofilm had no influence, too. The killing with increasing incubation time in NCT as a summary of the values of the weekly evaluation for each microorganism is depicted in Figure 7. The detection limit for the yeast could be nearly achieved after 4 h in NCT, while it was 1–2 h for bacteria.

### 3.4. Scanning Electron Microscopy

Analyzing the isolated biofilm cultures, we observed that all three strains showed typical biofilm morphology after 1 week of incubation if not treated with NCT. *K. variicola* biofilms were compact, covered with slime-like and protein-like substance binding between the bacteria. The *P. aeruginosa* and *S. aureus* biofilms, likewise, showed a three-dimensional structure although presenting less slime-like substance on their surface. Protein-like structures could be observed connecting the *P. aeruginosa* rods. The *S. aureus* biofilms showed a protein-like substance between and underneath the bacteria. After 30 min of treatment with NCT, the typical biofilm morphology of a 3-dimentional structure and the presence of slime-like substance was absent for *K. variicola*. The protein-like binding structures looked damaged in the *P. aeruginosa* and *S. aureus* biofilms, after 30 min of NCT. *K. variicola* and *P. arguginosa*, after 30 min of treatment, showed a damaged shape with impressions. *S. aureus* biofilms showed structural destruction in part of the bacteria. Similar morphological alterations could be observed for biofilms of all three strains after exposition to NCT for 60 min (Figure 8).

The 14-week-old control biofilms of *K. variicola*, *P. aeruginosa*, and *S. aureus* demonstrated typical mature biofilm morphology. All three strains showed clumps with the presence of water channels and slime-like structures covering and connecting the bacteria. These control biofilms presented morphological cell integrity in all strains. After 30 min of treatment with NCT, the biofilms showed less slime-like structures in all strains, although structural damage could be observed only in the *K. variicola* biofilms. After 60 min of treatment with NCT, all the strains presented morphological damage and absence of slime-like structures (Figure 9).

The 3-week-old control biofilms of *C. albicans* showed agglomerates of cells. The typical *C. albicans* biofilm morphology could not be observed, but cell morphology was intact. After 60 min treatment with NCT, cell clumps were not visible, and structural cell damage could be observed for all three strains tested. The same could be observed after 120 min of treatment with NCT (Figure 10).

The mixed long-term culture of *C. albicans* with *K. variicola*, *P. aeruginosa*, and *S. aureus*, after 3 weeks of incubation, showed typical biofilm structure for all strains in the control groups. *C. albicans* cultivated with *K. variicola* showed a homogeneous and three-dimensional biofilm formed by both microorganisms. In mixed culture, *C. albicans* seemed to predominate over *P. aeruginosa*. The culture of *C. albicans* together with *S. aureus* showed the bacteria growing attached to the surface of *C. albicans* hyphae. After 60 min of treatment with NCT, the *C. albicans* and *K. variicola* biofilms showed less volume and morphological alterations. The *C. albicans* plus *P. aeruginosa* and *C. albicans* plus *S. aureus* biofilms were completely eliminated from the surface of the metal plates after 60 min of NCT activity. The same finding was obtained for the mixed cultures treated for 120 min with NCT (Figure 11).

## 4. Discussion

In chronic infections and on foreign bodies, biofilms develop and may persist and be difficult to eradicate if they cannot be removed surgically or by replacement of the foreign body. Particularly in such cases, effective antimicrobial agents are desirable for treatment. Because of their good tolerability and efficacy in infected chronic skin ulcerations, according to a phase II clinical study and additional cases [19,27,28], and NCT may become a valuable therapeutic tool for such patients. It has shown microbicidal activity against bacterial biofilms, which were up to one week old [16,17,18]. Surprisingly, against *Staphylococcus epidermidis*, this activity appeared to be rather higher in biofilms grown for 7 days, compared with 1 day, without a clear explanation [16]. This might have been specific to the system or to NCT since, in general, mature biofilms seem to be more resistant against antiseptics. According to Stojicic and colleagues, after growing multispecies biofilms on collagen-coated hydroxyapatite disks, bacteria in mature biofilms were more resistant to 1% NaOCl, 0.2/0.4% iodine-potassium iodide, and 2% chlorhexidine than bacteria in young biofilms [22]. In this study, the biofilms were grown for 8 weeks, and a change in the susceptibility was found after 2–3 weeks [22], although biofilms are considered to have reached their last stage of maturation around 9–12 days after the beginning of their formation [22,29].

Considering all these previous knowledges, we wanted to clarify the effect of NCT against long-lasting biofilms over several weeks to months. Both of the devices we used for rapid development and persistence of biofilms, the MBEC^TM^ inoculator and metal discs, turned out to be suitable for this purpose. Metal discs are cheap, but not always readily available, while MBEC^TM^ inoculators are easier to handle, but more expensive, and their plastic surface was different from the materials used for implants in orthopedics. In the meantime, however, more surface types are available.

As a clear result, the susceptibility of bacterial biofilms to NCT did not change with their age up to 3.5 months and more. The same was true for yeast biofilms and mixed biofilms (bacteria and yeasts) grown for at least one month. As in scanning electron microscopy, the biofilm developed after one week and persisted with some possible increase in thickness but without marked morphological changes on both devices. This is in accordance with the previous findings of maturation after 1–2 weeks [22,29] and may be the reason why the susceptibility of the microorganisms to NCT did not change with the age of the biofilms. In a previous study, a biofilm of *Staphylococcus epidermidis* grown for 7 days was rather more susceptible to NCT than a biofilm grown for 2 days [16]. Longer periods were not tested. In the same study, the biofilm of *S. epidermidis* producing poly-N-acetylglucosamine, a main compound for the extracellular matrix, was more resistant to NCT compared with a mutant strain not producing this substance [16]. This indicates that metabolic changes of a biofilm actually can influence its susceptibility to NCT to a limited degree, but does not cause resistance. About the metabolism of the biofilms in the present study, however, no statements are possible since we did not perform functional tests. It can be said that at least no metabolic changes with a significant influence on the susceptibility to NCT occurred.

As it is generally the case with biofilms, their resistance to antiseptics and antimicrobial agents usually is higher than that of planktonic cultures due to hindered penetration and to the heterogeneity of the population of single microorganisms. The same is true for NCT, whereby the present study confirms two previous ones on short-term biofilms [16,17]. The sequence of lower to higher susceptibility of the tested strains turned out to be similar for planktonic and biofilm cultures, with *C. albicans* > *S. aureus* > *P. aeruginosa* > *K. variicola* (for planktonic cultures for bacteria and yeasts, see for instance [11,30,31]). *S. aureus* and *S. epidermidis* needed longer incubation times than *P. aeruginosa* in a previous biofilm study, too [17]. Incubation times required for killing are determined by penetration of NCT into the microorganisms and therefore by their different coatings [32]. Obviously, penetration is the decisive step in biofilms. Destruction and loss of viability in the presence of NCT at a pH of around 7 and at 37 °C needs more time in biofilms, 0.5–1 h in bacteria and 3–4 h in *C. albicans*, than against planktonic suspensions (approximately a few—20 min in bacteria, 1–2 h in *C. albicans*). Killing tests and electron microscopy seem to correlate in the present study.

These times required for inactivation of biofilms by NCT in vitro may appear as relatively long for successful application in vivo. In this regard, however, it is important that the microbicidal activity of NCT is not decreased but increased in the presence of human body fluids and exudates [11,15,33,34,35]. The reason is the transfer of the active chlorine of NCT to other amino compounds and the formation of corresponding chloramines in equilibrium (transchlorination) [36]. Some of them are more lipophilic than NCT, particularly monochloramine (NH_2_Cl), and penetrate and kill microorganisms faster [11,15,33,34,35]. The efficacy of NCT in chronic wound infections may also indicate that transchlorination plays a significant role in the attack of biofilms by NCT [19,27,28].

Against mixed biofilms, NCT can be an interesting agent as well. The interaction of microbial species in multispecies biofilms leads to the development of antibiotic resistance as well as protective mechanisms against host immune response [37]. Frequently, the interaction of *C. albicans* and bacteria, either physically or chemically, appears to be causative [38,39]. Physical interaction is achieved through the assistance of cell surface receptors/proteins [40]. By contrast, chemical interactions occur due to the involvement of several secretory molecules or metabolic intermediates released either by fungi or bacteria into the environment. These metabolic products/or secretory molecules facilitate communication between fungal and bacterial cells, either synergistically or antagonistically [41]. As an active chlorine compound, NCT has been shown to oxidize and chlorinate secreted virulence factors of bacteria and fungi and to inactivate these factors, as seen in shiga toxin of *Escherichia coli* [42], staphylococcal toxins [43], gliotoxin of *Aspergillus fumigatus* [44], and aspartyl proteinases of *Candida albicans* [31]. A recent comprehensive investigation disclosed a decrease in a panel of virulence factors in NCT-treated A. fumigatus [45]. Therefore, it is well conceivable that NCT blocks communication and interaction factors of mixed species in biofilms as well. Actually, NCT was equally effective against mixed biofilms compared to the single-species biofilms in the present study. The yeast was killed later than the bacteria, which is explainable by slower penetration into the fungus, as mentioned above. Because of the unspecific oxidizing mechanism of action with multiple targets [46], resistance against NCT and other active chlorine compounds in therapeutic concentration is hardly imaginable and has never been observed [47]. This underlines the general advantage of antiseptics versus antibiotics for topical treatment of infections and can be confirmed by the successful application of NCT in therapy-refractory wound infections [27,28].

## 5. Conclusions

NCT exerts constant bactericidal and fungicidal activity in biofilms grown for weeks to months. This is also true for biofilms containing a combination of bacteria and yeasts. Therefore, NCT is a promising medication for therapy of infections where long-term biofilms play a role, for instance, infected chronic wounds, which is supported by previous clinical results.

## Figures and Tables

**Figure 1 antibiotics-10-00891-f001:**
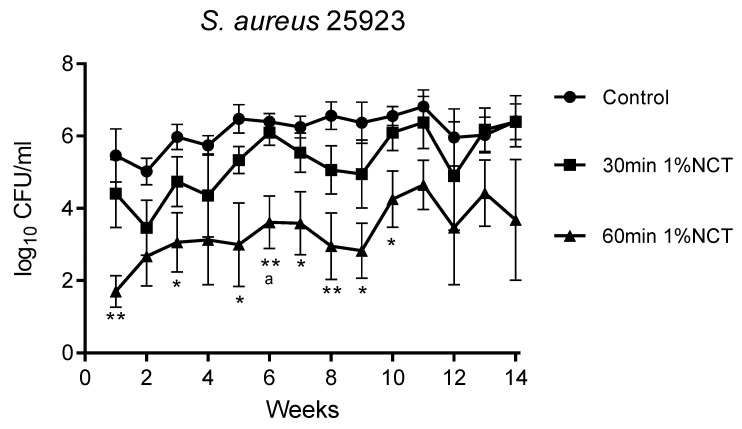
Bactericidal activity of 1% NCT at pH 7 and 37 °C against biofilms of *S. aureus* ATCC 25923 grown for 14 weeks. Weekly evaluation by quantitative killing assays. Mean values ± SEM, *n* = 5 (*n* = 3 for weeks 12–14); * *p* < 0.05 versus control; ** *p* < 0.01 versus control; ^a^ *p* < 0.01 between 30 and 60 min NCT.

**Figure 2 antibiotics-10-00891-f002:**
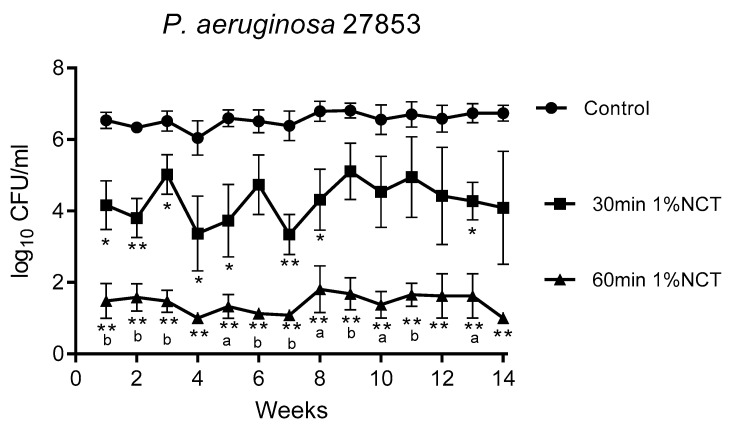
Bactericidal activity of 1% NCT at pH 7 and 37 °C against biofilms of *P. aeruginosa* ATCC 27853 grown for 14 weeks. Weekly evaluation by quantitative killing assays. Mean values ± SEM, *n* = 6 (*n* = 4 for weeks 12 and 13, *n* = 3 for week 14); * *p* < 0.05 versus control; ** *p* < 0.01 versus control; ^a^ *p* < 0.05 versus 30 min NCT; ^b^ *p* < 0.01 versus 30 min NCT.

**Figure 3 antibiotics-10-00891-f003:**
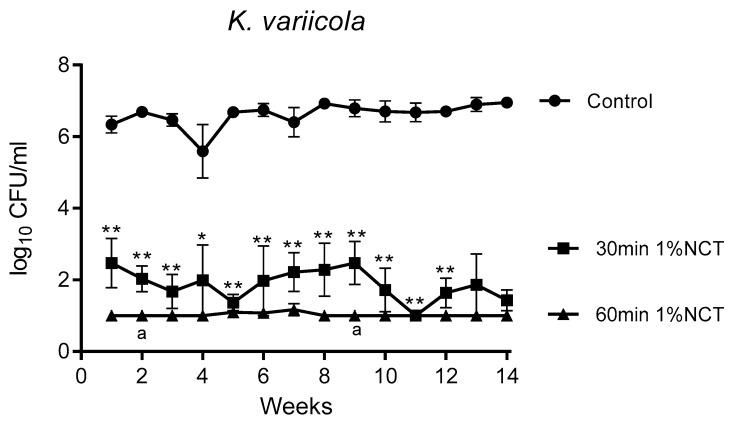
Bactericidal activity of 1% NCT at pH 7 and 37 °C against biofilms of *K. variicola* clinical isolate grown for 14 weeks. Weekly evaluation by quantitative killing assays. Mean values ± SEM, *n* = 5 (*n* = 4 for weeks 12–14); *p* < 0.01 for all values of 30 min and 60 min NCT versus control except for *p* < 0.05 for 30 min NCT after 4 weeks; * *p* < 0.05 versus control; ** *p* < 0.01 versus control; ^a^ *p* < 0.05 of 30 min versus 60 min NCT.

**Figure 4 antibiotics-10-00891-f004:**
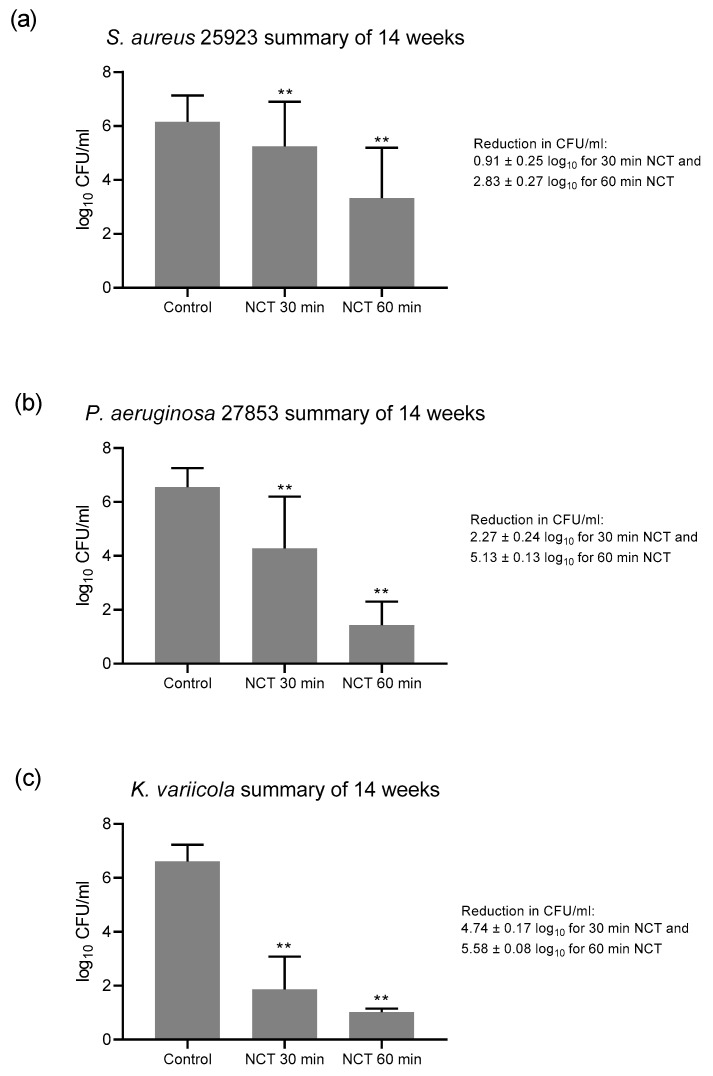
Summary of all values of NCT and controls over the 14 weeks of Figure 1, Figure 2 and Figure 3. (**a**) *S. aureus*, *n* = 62; (**b**) *P. aeruginosa*, *n* = 73–74; (**c**) *K. variicola*, *n* = 64–65. Mean values ± SD. ** *p* < 0.01 for controls versus NCT and for 30 min versus 60 min incubation time in all panels.

**Figure 5 antibiotics-10-00891-f005:**
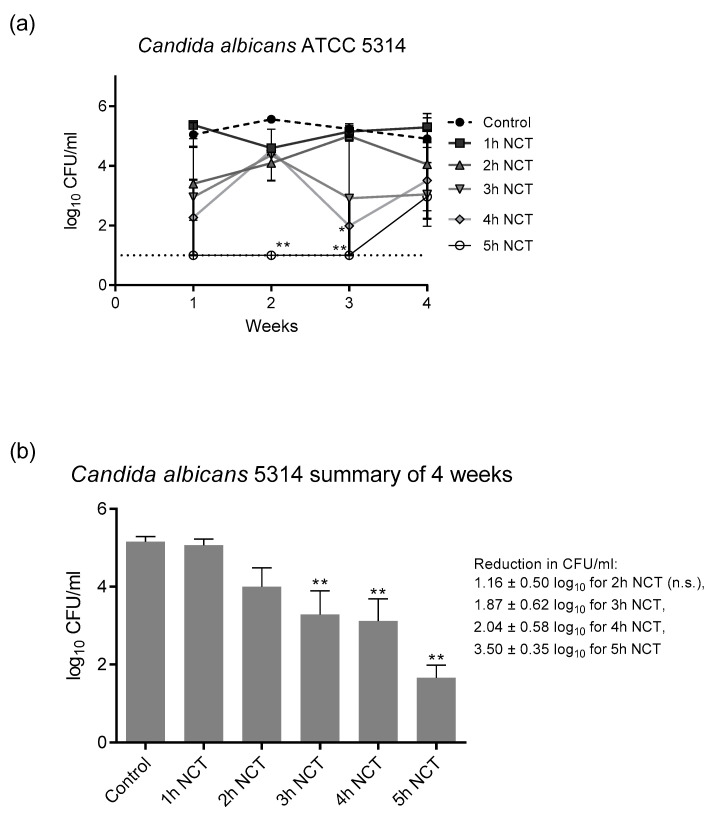
(**a**) Fungicidal activity of 1% NCT at pH 7 and 37 °C against biofilms of *C. albicans* ATCC 5314 grown for 4 weeks. Weekly evaluation by quantitative killing assays. Mean values ± SEM, *n* = 3 (*n* = 2 for week 2); * *p* < 0.05 versus control; ** *p* < 0.01 versus control. (**b**) Summary of all values of NCT and controls over the 4 weeks of Figure 5a. Mean values ± SEM, *n* = 9–12; ** *p* < 0.01 versus control; n.s.: not significant.

**Figure 6 antibiotics-10-00891-f006:**
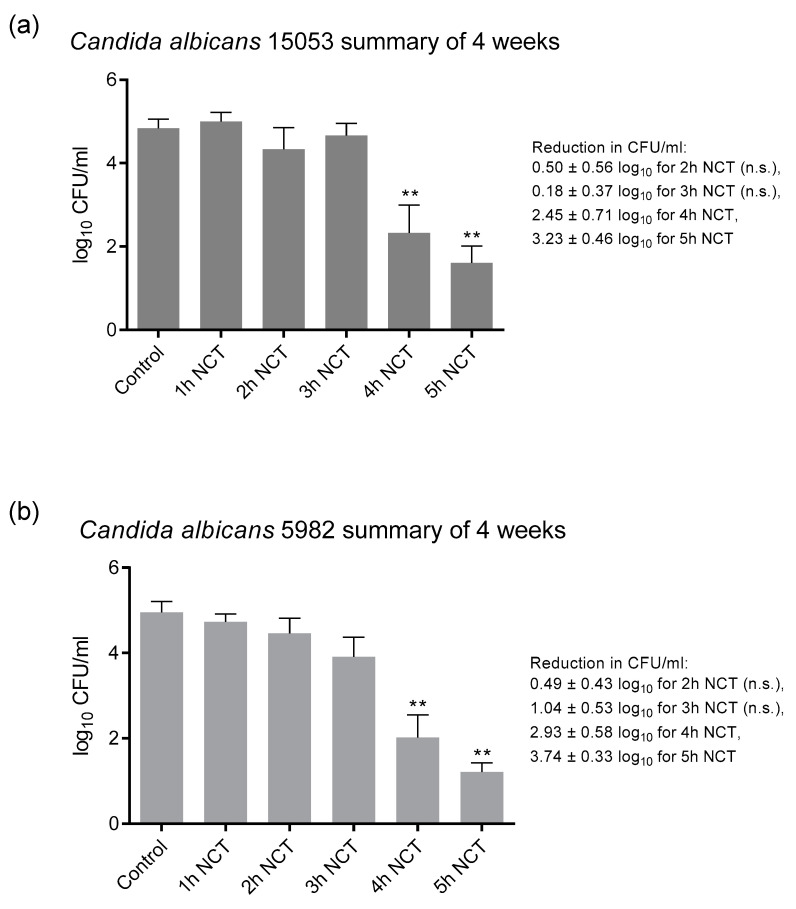
Fungicidal activity of 1% NCT at pH 7 and 37 °C against biofilms of *C. albicans* ATCC 15053 (**a**) and CBS 5982 (**b**) grown for 4 weeks. Weekly evaluation by quantitative killing assays. Summary of all values of NCT and controls. Mean values ± SEM, *n* = 9–12 values each from 2–3 independent experiments; ** *p* < 0.01 versus control; n.s.: not significant.

**Figure 7 antibiotics-10-00891-f007:**
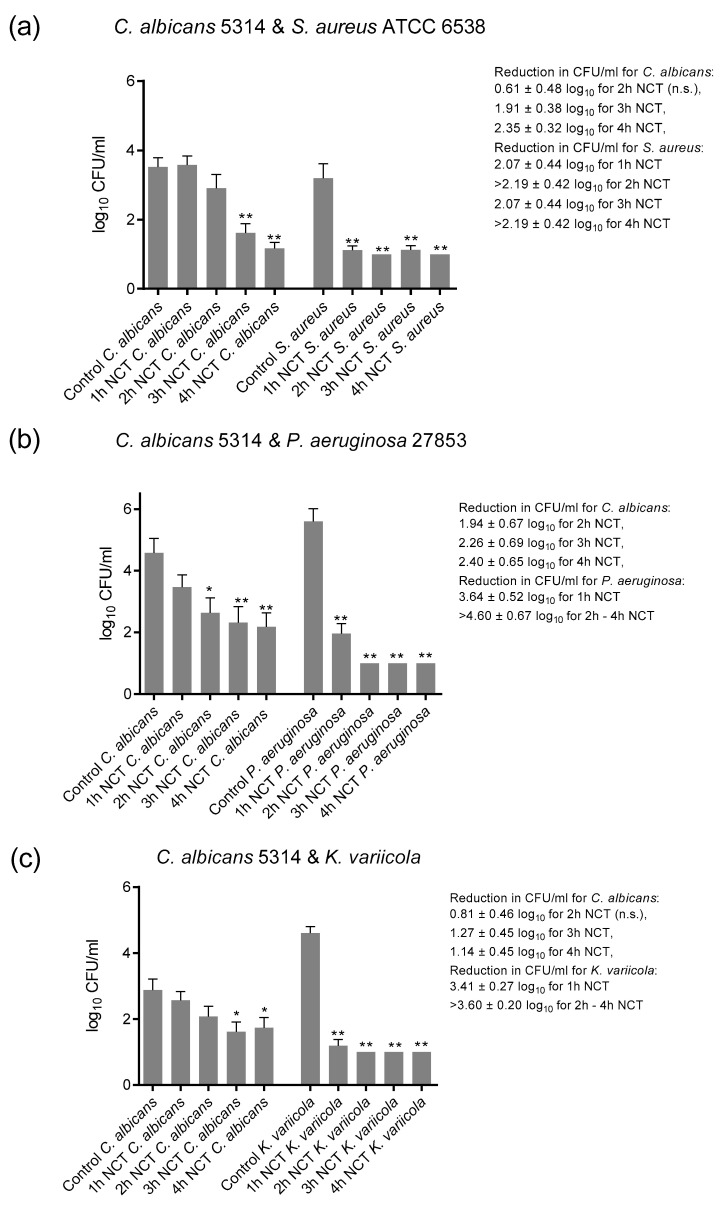
Fungicidal and bactericidal activity of 1% NCT at pH 7 and 37 °C against mixed biofilms of *C. albicans* ATCC 5314 plus *S. aureus* ATCC 6538 (**a**), *P. aeruginosa* ATCC 27853 (**b**) or *K. variicola* (**c**) clinical isolate grown for 5 (*S. aureus*) to 7 weeks. Weekly evaluation by quantitative killing assays. Summary of all values of NCT and controls. Mean values ± SEM, *n* = 15–17 (*S. aureus*), 10–15 (*P. aeruginosa*), and 18–20 (*K. variicola*) values each from 3–5 independent experiments; * *p* < 0.05 versus control; ** *p* < 0.01 versus control; n.s.: not significant.

**Figure 8 antibiotics-10-00891-f008:**
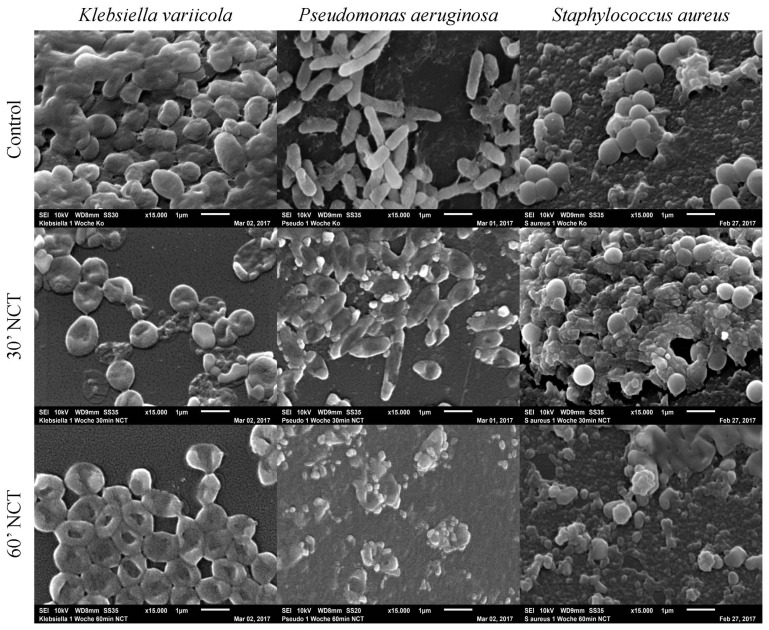
Scanning electron microscopy of 1-week-old Klebsiella variicola, Pseudomonas aeruginosa, and Staphylococcus au-reus biofilms. Upper row: mock-treated control biofilms (Magnification 15,000×); middle row: biofilms treated with NCT, for 30 min (Magnification 15,000×); lower row: biofilms treated with NCT, for 60 min (Magnification 15,000×). Specimens were analyzed by scanning electron microscopy (SEM, JSM-6010LV, JEOL GmbH, Freising, Germany).

**Figure 9 antibiotics-10-00891-f009:**
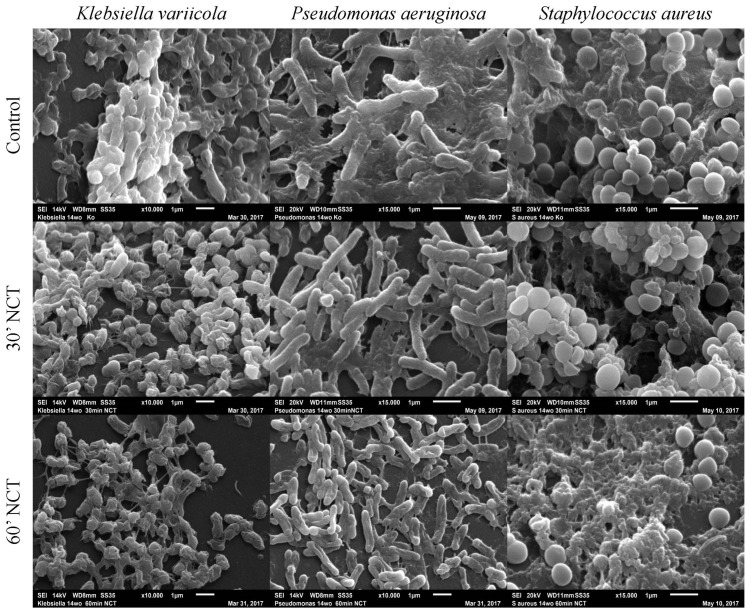
Scanning electron microscopy of 14-week-old *Klebsiella variicola*, *Pseudomonas aeruginosa*, and *Staphylococcus aureus* biofilms. Upper row: mock-treated control biofilms (Magnification: *K. variicola* 10,000×; *P. aeruginosa* and *S. aureus* 15,000×); middle row: biofilms treated with NCT, for 30 min (Magnification: *K. variicola* 10,000×; *P. aeruginosa* and *S. aureus* 15,000×); lower row: biofilms treated with NCT for 60 min (Magnification: *K. variicola* and *P. aeruginosa* 10,000×; *S. aureus* 15,000×). Specimens were analyzed by scanning electron microscopy (SEM, JSM-6010LV, JEOL GmbH, Freising, Germany).

**Figure 10 antibiotics-10-00891-f010:**
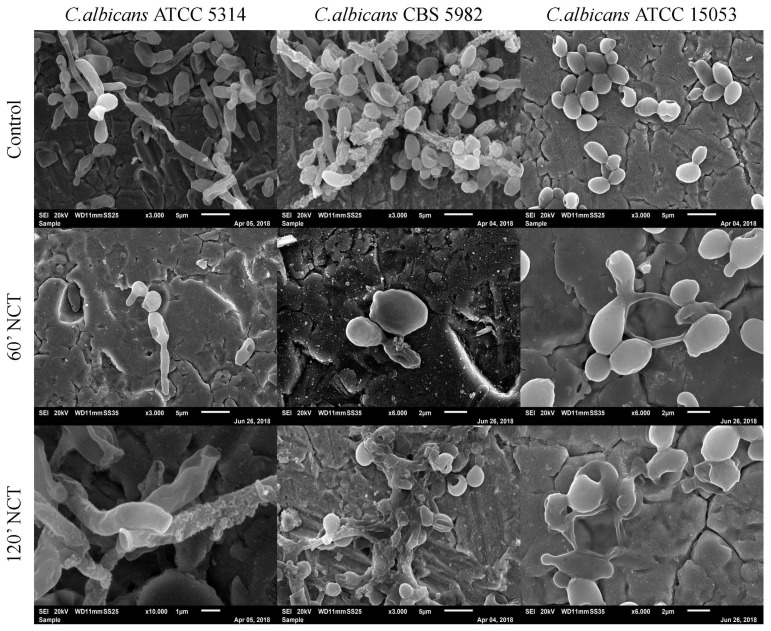
Scanning electron microscopy of 3-week-old *C. albicans* ATCC 5314, *C. albicans* CBS 5982 and *C. albicans* ATCC 15053 biofilms. Upper row: mock-treated control biofilms (Magnification: 3000×); middle row: biofilms treated with NCT, for 60 min (Magnification: *C. albicans* ATCC 5314 3000×; *C. albicans* CBS 5982 and *C. albicans* ATCC 15053 6000×); lower row: biofilms treated with NCT for 120 min (Magnification: *C. albicans* ATCC 5314 10,000×; *C. albicans* CBS 5982 3000× and *C. albicans* ATCC 15053 6000×). Specimens were analyzed by scanning electron microscopy (SEM, JSM-6010LV, JEOL GmbH, Freising, Germany).

**Figure 11 antibiotics-10-00891-f011:**
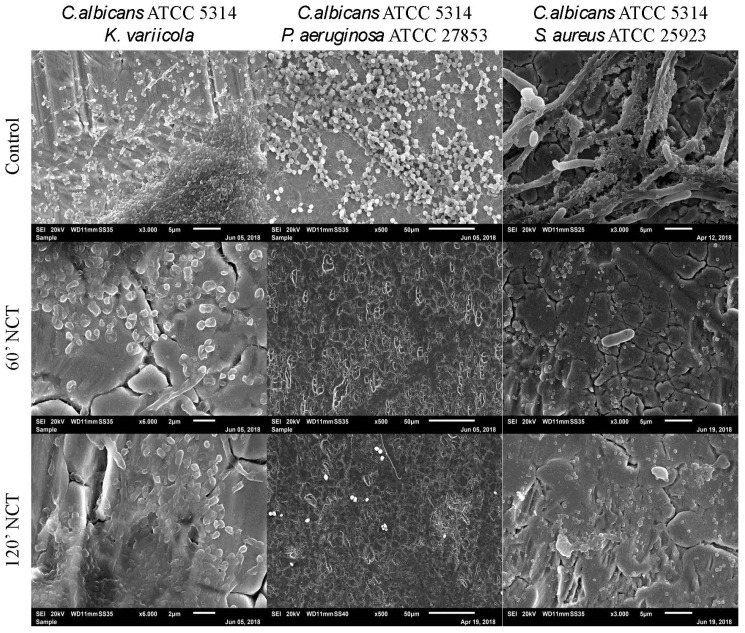
Scanning electron microscopy of 3-week-old mixed biofilms using *C. albicans* co-cultured with *K. variicola, P. aeruginosa* or *S. aureus*. Upper row: mock-treated control biofilms (Magnification: *C. albicans* plus *K. variicola* 3000×; *C. albicans* plus *P. aeruginosa* 500×; *C. albicans* plus *S. aureus* 3000×); middle row: biofilms treated with NCT, for 60 min (Magnification: *C. albicans* plus *K. variicola* 6000×; *C. albicans* plus *P. aeruginosa* 500×; *C. albicans* plus *S. aureus* 3000×); lower row: biofilms treated with NCT, for 120 min (Magnification: *C. albicans* plus *K. variicola* 6000×; *C. albicans* plus *P. aeruginosa* 500×; *C. albicans* plus *S. aureus* 3000×). Specimens were analyzed by scanning electron microscopy (SEM, JSM-6010LV, JEOL GmbH, Freising, Germany).

## Data Availability

All data are presented in the publication.

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
