# Peer review of "Activity of N-Chlorotaurine against Long-Term Biofilms of Bacteria and Yeasts"

_antibiotics, 2021, doi:10.3390/antibiotics10080891_

Round 1
Reviewer 1 Report
This is an interesting manuscript about the effect of the activity of NCT against bacteria and yeasts in longer-lasting biofilms, including their co-culture. In this manuscript, Staphylococcus aureus, Pseudomonas aeruginosa, and Klebsiella variicola biofilms were grown for 14 weeks in MBECTM inoculator with 96 well base. Similar tests were done with C. albicans biofilms.
Viable counts of biofilms of single bacteria were reduced after treatments in Gram-negative bacteria biofilm in C. albicans biofilm and in combined ones. These results can explain the clinical efficacy of NCT in infected chronic wounds.
The manuscript is well written and after some minor revision could be suitable for publication. Although the overall conclusion reached by the authors is reasonable, as currently presented, this manuscript has a number of deficiencies that need to be addressed, outlined below:
There are numerous errors of “Tense and Grammar” throughout the manuscript. Therefore, a diligent editing is in order to fix the ENGLISH language.
Introduction
Line 37: Please verify the presence of a verb in the principal sentence.
Line 38: Please consider writing “ These infections increase the pressure on medical systems already overwhelmed in both developed and undeveloped countries” instead of the present sentence.
Line 40: Please consider writing “Because of the rising of number of patients with this serious complication continues to rise” instead of the present sentence.
Line 43 Consider writing “isolated microorganisms” instead of “ cultered microorganisms”.
Line 45: Please explain better this sentence “In previous decades, commercially available antibiotics lost its activity against CoNS and several other microorganisms. “
Line 49 Consider writing “ability” instead of “ potential”.
Line 53 remove the sentence " Biofilms are a long-lived, organized community of microorganisms within an extracellular matrix. ", which is not useful here.
Line 56-58 Consider explaining the sentence “anti-infective for topical treatment of infections of different body sites”.
Line 59-61 Please consider rewriting this sentence.
Line 61 should be COVID-19 with uppercase letters.
Line 69 Not clear, please rewriting the sentence “and reduction of the viability of dental plaque was found”.
Line 71 Consider writing “started to focus on” instead of “ became interested in”.
Materials and methods
Line 78 : Please explain what this mean “really deep frozen for storage”
Line 81: should be CFU with uppercase letters.
Line 87 remove the word sensitive because it is not useful
Line 102 Consider writing “The initial concentration of bacteria is” instead of “The number of bacteria came to 3-5 x 107 colony forming units (cfu) / ml at the start.”
Line 102 should be CFU with uppercase letters.
Line Consider writing “The initial concentration of yeast is” instead of “The number of yeast came to approximately 1 x 121 105 colony forming units (cfu) / ml at the start.”
Line 151 should be CFU with uppercase letters.
Results
Line 179 should be CFU with uppercase letters.
Line 192 should be CFU with uppercase letters.
Figure 1,2,3,4, should be CFU with uppercase letters.
Line 215-216 should be CFU with uppercase letters.
Figure 5-6 should be CFU with uppercase letters.
Discussion
Line 323 Do not understand the meaning of this sentence “or by change of the foreign body.”
Line 363 Consider writing “usually” instead of “in general”.
Author Response
Answers to Reviewer 1:
We are grateful to the reviewer for the endeavours and the valuable suggestions. Hopefully, we could address all points satisfactorily.
Concern:
There are numerous errors of “Tense and Grammar” throughout the manuscript. Therefore, a diligent editing is in order to fix the ENGLISH language.
Answer:
We have addressed all points according to the suggestions of the reviewer. Please see the marked sentences in the revised manuscript.
Only one point of the reviewer was unclear to us:
Concern:
Line 56-58 Consider explaining the sentence “anti-infective for topical treatment of infections of different body sites”.
Answer:
In our opinion, the sentence is clear. Detailed informations are provided in the cited references. If there is any special information we should provide, please advice.
Reviewer 2 Report
The authors have evaluated the effect of N-chlorotaurine on aged biofilms formed by bacteria, yeast and mixed yeast-bacteria. The biofilms were grown on MDEC pegs or metal discs. They show that the biofilms susceptibility to NCT was constant and not dependent of the age of the biofilm. The results are clearly presented.
There is a minor point which could be addressed. Why results are presented in Figure 4 as mean values +/- SD while in all other figures, presenting the same kind of results, they are shown as mean +/- SEM?
Author Response
Answers to Reviewer 2
We are grateful to the reviewer for the endeavours. Hopefully, we could explain the single point satisfactorily.
Concern:
There is a minor point which could be addressed. Why results are presented in Figure 4 as mean values +/- SD while in all other figures, presenting the same kind of results, they are shown as mean +/- SEM?
Answer and explanation:
In Figure 1-3, we applied SEM since the use of SD would have caused an overlay of deviation bars, which would have made the figures difficult to understand.
In Figure 4, however, the usual application of SD is better since the application of SEM would have caused minimal error bars because of the high number of n.
Reviewer 3 Report
Dear Authors,
It is to be appreciated all the research activity presented in this work, the results are good, but not novelty. In other words, the novelty is not highlighted. There are many studies with similar results, including those of the authors.
Also, more details for future applications should be provided.
Author Response
Reviewer 3
We are grateful to the reviewer for the endeavours and the valuable suggestions. Hopefully, we could address all points satisfactorily.
Concern:
It is to be appreciated all the research activity presented in this work, the results are good, but not novelty. In other words, the novelty is not highlighted. There are many studies with similar results, including those of the authors.
Answer:
It is true that we have published many studies on the microbicidal activity of NCT and there are three studies (two from us) on its anti-biofilm activity. The activity of NCT against long-term biofilm, however, has not yet been investigated and this is new here as well as the activity against a combined biofilm.
These aspects have been more stressed in the manuscript in lines 71-80.
Concern:
Also, more details for future applications should be provided.
Answer:
More details have been provided in lines 58-62.
Many aspects can be read in our reviews and original papers, respectively, which are mentioned in references 11-15 and 27-28, for instance.
Reviewer 4 Report
The Authors in this paper carried out further studies on the natural compound N-chlorotaurine (NCT). Such a compound were tested for antibiofilm properties against polymicrobial biofilms (fungi and bacteria mixed) after log times of incubation (5-7 weeks). NCT resulted active against long-term biofilms and combined biofilms cultures of yeasts and bacteria showed a similar susceptibility pattern to NCT as single ones.
This study added important informations about NCT, that is already under clinical evaluation, for a therapeutic application.
MInor changes.
- The Authors did not explain because Klebsiella variicola was included in the study.
- In the Introduction section Authors should clarify the importance to test NCT at very long time of incubation.
- Fig 1, 2 and 3 should be merged in a single figure
- In Figure 4 analysis statistics is missing
Author Response
Reviewer 4
We are grateful to the reviewer for the endeavours and the valuable suggestions. Hopefully, we could address all points satisfactorily.
Minor changes.
- The Authors did not explain because Klebsiella variicola was included in the study.
Answer:
We chose this species as an example for an emerging opportunistic pathogen. This has been added in lines 85-88.
- In the Introduction section Authors should clarify the importance to test NCT at very long time of incubation.
Answer:
NCT showed clinical efficacy in diseases where persisting biofilms appear to play a role for therapeutic problems. However, the activity of NCT against such biofilms was unknown. These considerations have been added in lines 71-75.
- Fig 1, 2 and 3 should be merged in a single figure
Answer:
We understand the idea of the reviewer very well. There are, however, specific differences in the number of experiments and in the statistical description (P values). Therefore, a single legend for all three figures seems to become confusing for the readers, while a presentation in 3 figures improves clarity. For this reason, we decided for the latter version.
- In Figure 4 analysis statistics is missing
Answer:
Statistics have been described in the legend:
“P < 0.01 for controls versus NCT and for 30 min versus 60 min incubation time in all panels.“
Since this can be overlooked, we have added stars in the panels as suggested. In addition, we changed the order in the legend to render it easier for the reader.